# Prenatal Alcohol Exposure and the Facial Phenotype in Adolescents: A Study Based on Meconium Ethyl Glucuronide

**DOI:** 10.3390/brainsci11020154

**Published:** 2021-01-25

**Authors:** Janina Maschke, Jakob Roetner, Tamme W. Goecke, Peter A. Fasching, Matthias W. Beckmann, Oliver Kratz, Gunther H. Moll, Bernd Lenz, Johannes Kornhuber, Anna Eichler

**Affiliations:** 1Department of Child and Adolescent Mental Health, University Hospital Erlangen, Friedrich-Alexander University Erlangen-Nürnberg, 91054 Erlangen, Germany; Jakob.Roetner@uk-erlangen.de (J.R.); Oliver.Kratz@uk-erlangen.de (O.K.); Gunther.Moll@uk-erlangen.de (G.H.M.); Anna.Eichler@uk-erlangen.de (A.E.); 2Department of Obstetrics and Gynecology, University Hospital Erlangen, Friedrich-Alexander University Erlangen-Nürnberg, 91054 Erlangen, Germany; tamme.goecke@ro-med.de (T.W.G.); Peter.Fasching@uk-erlangen.de (P.A.F.); Matthias.Beckmann@uk-erlangen.de (M.W.B.); 3Department of Gynecology, RoMed Klinikum Rosenheim, 83022 Rosenheim, Germany; 4Department of Psychiatry and Psychotherapy, University Hospital Erlangen, Friedrich-Alexander University Erlangen-Nürnberg, 91054 Erlangen, Germany; bernd.lenz@zi-mannheim.de (B.L.); Johannes.Kornhuber@uk-erlangen.de (J.K.); 5Department of Addictive Behavior and Addiction Medicine, Central Institute of Mental Health (CIMH), Medical Faculty Mannheim, Heidelberg University, 68159 Mannheim, Germany

**Keywords:** facial malformations, intrauterine alcohol exposure, ethyl glucuronide (EtG), meconium, alcohol, craniofacial shape, biomarker, palpebral fissure length, inner canthal distance, lip circularity

## Abstract

Here, we explore the effects of prenatal alcohol exposure (PAE) in adolescence. We investigated associations between meconium ethyl glucoronide (EtG) and facial malformation. For 129 children (66/63 male/female; M = 13.3, SD = 0.32, 12–14 years), PAE was implemented by newborn meconium EtG and maternal self-reports during the third trimester. Cognitive development was operationalized by standardized scores (WISC V). The EtG cut-off values were set at ≥10 ng/g (*n* = 32, 24.8% EtG10+) and ≥112 ng/g (*n* = 20, 15.5% EtG112+). The craniofacial shape was measured using FAS Facial Photographic Analysis Software. EtG10+− and EtG112+-affected children exhibited a shorter palpebral fissure length (*p =* 0.031/*p* = 0.055). Lip circularity was smaller in EtG112+-affected children (*p* = 0.026). Maternal self-reports were not associated (*p* > 0.164). Lip circularity correlated with fluid reasoning (EtG10+ *p =* 0.031; EtG112+ *p* = 0.298) and working memory (EtG10+ *p* = 0.084; EtG112+ *p* = 0.144). The present study demonstrates visible effects of the facial phenotype in exposed adolescents. Facial malformation was associated with a child’s cognitive performance in the alcohol-exposed group. The EtG biomarker was a better predictor than maternal self-reports.

## 1. Introduction

Prenatal alcohol exposure (PAE) is a known risk factor for child development and can cause adverse and irreversible damage to the child [1]. Alcohol consumption is one of the major preventable causes of childhood health and developmental problems and continues to be a public health challenge [2,3]. The prevalence of self-reported alcohol use in pregnancy is about 10% worldwide [4]. Further research has reported a 20% prevalence of alcohol consumption during pregnancy in European countries [5,6]. The prevalence of fetal alcohol syndrome (FAS) in a general population has been reported to be 1.1 per 1000 (0.11%), and 7.7 per 1000 (0.77%) for fetal alcohol spectrum disorder (FASD) [4,7]. Alcohol, as a teratogenic substance, can damage brain development throughout pregnancy and cause structural abnormalities (i.e., facial malformation) or affect the brain at the neuro-biochemical level in the form of brain dysfunction with different consequences [8]. This means that PAE has structural and functional consequences for the central nervous system (CNS). A global decrease in intelligence or developmental delay is summarized by the FASD diagnosis criterion ‘CNS abnormality’. There is a homogeneous reduction in action and verbal IQ [9,10]. In addition to the impairment of global intelligence, PAE might impact child learning and memory abilities. The reduced memory function affects the short- and long-term memory and is linked to slower information processing and a reduced learning capacity [11,12,13]. Further performance deviations relate to language abilities, fine motor skills, spatial-visual perception, executive functions, arithmetic skills, awareness, and social skills [14,15]. Current research based on ethanol metabolites in the meconium found that even small amounts of these metabolites significantly influence cognitive development in childhood and adolescence [16,17,18].

The duration of alcohol consumption during pregnancy and the timing of prenatal alcohol exposure differ from mother to mother, and numerous other developmental risk factors are significant. Therefore, the negative effects vary greatly between children with FASD, and each child is affected differently [13]. A recent study on FASD children revealed that all affected mothers drunk before pregnancy recognition, but only 10% continued alcohol consumption during the second and third trimester [19]. Therefore, there is no safe-time-point or safe-amount for prenatal alcohol consumption: The range from ‘visible’ to ‘invisible’ effects caused by smaller or higher amounts of prenatally consumed alcohol is wide. The basis for valid studies on the developmental consequences of intrauterine alcohol exposure is reliable alcohol-exposure measures. Most studies rely on a clinical FAS diagnosis or maternal self-reports. However, it has been shown that maternal self-reports are inaccurate compared to biomarkers [20,21]. Other studies suggest that metabolites could be a more effective marker than the maternal self-report for predicting negative development outcomes [16,20,22].

For FAS diagnosis, in addition to CNS and growth abnormalities, craniofacial changes are used as physical diagnostic criteria [2,23,24,25,26,27]. Animal studies have reported that the facial shape is formed up to day 16 of pregnancy [28], making facial malformations an early marker of intrauterine alcohol exposure. A short palpebral fissure (PFL), a weakly modeled or flat philtrum, and a thin upper lip are the most frequently observed characteristics for alcohol embryopathy and children suffering from FASD (see Figure 1); these typical facial features disappear with an increasing age and become less clear over time [14,29]. Muggli [2] and Tan [23] found that low to moderate PAE (operationalized by maternal self-reports) in 1- and 15-month-old children influences the craniofacial shape. Muggli [2] observed that the craniofacial differences concentrated around the midface, especially the nose, eyes, and lips [2].Tan [23] observed similar structural changes in the face, including a smaller forehead, a larger maxilla on one side of the face, and a short and upturned nose [25]. In clinical settings, professionals use the Lip Philtrum Guide to observe the upper lip size and the philtrum. A sliding digital caliper or ruler is used to examine the palpebral fissure length (PFL). However, these manual measurements are error-prone [29,30,31,32]. To address this, Astley and Clarren [30] developed a software tool (the FAS Facial Photographic Analysis Software) to examine facial changes in children. The software was developed for research and health care professionals [32].

It can be summarized that the facial phenotype changes due to early intrauterine alcohol exposure; these changes have a high interindividual variance—in addition to others, depending on the drinking amount and timing—which cannot be accurately measured by maternal self-reports because of invalid reporting behavior. We are aware of two published studies that have associated self-report operationalized alcohol exposure with facial anomalies in infants [2,26]. We are aware of no studies using biomarkers to measure alcohol exposure or studies analyzing facial differences in adolescence.

The present study examined the association between PAE, operationalized by an ethanol metabolite in newborn meconium (EtG), and craniofacial differences in 12 to 14-year-old adolescents. These associations were compared with associations between the maternal self-report on PAE and craniofacial shapes. We also analyzed the functional relevance of detected facial differences in fluid reasoning and working memory cognitive performance.

## 2. Materials and Methods

### 2.1. Study Design

The study is a cooperation project between the Department of Obstetrics and Gynecology, Psychiatry and Psychotherapy, and Child and Adolescent Mental Health at the University Hospital Erlangen. The first part of the study (Franconian Maternal Health Evaluation Studies (FRAMES)) was performed at the Department of Obstetrics and Gynecology and took place from 2005 to 2007; 1100 women were recruited in the third trimester of their pregnancy [33,34]. From 2012 to 2015, a random subsample of *n* = 501 was contacted for re-participation. Since the risk groups (prenatal alcohol consumption and/or prenatal depression) were underrepresented in this sample, *n* = 117 additional women were contacted (oversampling). Finally, *n* = 245 FRAMES mothers and their children (39.6%; child age: M = 7.74, SD = 0.74, range: 6.00–10.0) took part in the FRANCES I study (Franconian Cognition and Emotion Studies) at the Department of Child and Adolescent Mental Health. The 245 participating women did not differ from the 373 non-participating women at the time of birth (FRAMES) in terms of the family status (χ^2^(1) = 0.16, *p* = 0.690), school level (χ^2^(1) = 0.08, *p* = 0.774), or family income (χ^2^(2) = 0.97, *p* = 0.616). From 2019, the mothers and their adolescent children were contacted again to ask them to take part in a second follow-up (FRANCES II). The recruiting is ongoing. For the present paper, there were *n* = 185 re-participants (75.5%). The re-participating women did not differ from the *n* = 60 non-re-participating women in terms of the family status (χ^2^(1) = 1.43, *p* = 0.232) or family income (χ^2^(2) = 0.62, *p* = 0.735) at the time of birth (FRAMES). However, their school level was higher (χ^2^(1) = 7.96, *p* = 0.005).

In the FRANCES I and II follow-up study, multiple parameters were examined in a multi-level-design (clinical, neuropsychological, neurophysiological, and neurobiological outcomes) observing the cognitive, emotional, and social development. In the FRANCES II follow-up, the facial malformations of young adolescents were measured. The study was approved by the Ethik-Kommission from the Friedrich-Alexander Universität Erlangen—Nürnberg (FAU) (353_18B) and conducted in accordance with the Declaration of Helsinki. Mothers and children gave approved consent. Participation in the study was voluntary.

### 2.2. Sample Characteristics

The participants were between the ages of 12 and 14 years (M = 13.3, SD = 0.32, range 12 to 14 years). The overall median school type was 4 (1 = primary degree (i.e., ‘Förderschule’), and school ranges were 2 = secondary degree (‘Hauptschule’), 3 = intermediary degree (‘Realschule’), and 4 = upper secondary degree (‘Gymnasium’)) [35]. Photographs of *n* = 165 participants were taken. From these (*n* = 165) participants, *n* = 36 children had to be excluded: *n* = 29 children had no valid EtG measurements; *n* = 3 children were outliers regarding >4 SD around the mean; *n* = 1 was excluded due to missing the sticker for the photograph; and *n* = 3 children with one parent being non-Caucasian (China, Nigeria, and Peru) (these children were excluded because of established ethnic differences in facial abnormalities due to PAE). Consequently, the present paper reports the data of 129 children who had valid EtG measurements and for whom facial photographs were taken. Sample characteristics are reported in Table 1. Out of the 129 children, *n* = 32 children (24.8%) had a meconium EtG ≥ 10 ng/g (EtG10+) and *n* = 20 children (15.5%) had a meconium EtG ≥ 112 ng/g (EtG112+). The EtG+ (≥10 ng/g) children were compared to the non-exposed children within the sample (EtG negative: EtG10− *n* = 97, EtG112− *n* = 109). For maternal self-report, there were *n* = 32 mothers (24.8%) who reported alcohol consumption during pregnancy.

### 2.3. Instruments and Measures

Prenatal alcohol consumption: Meconium EtG is an ethanol metabolite that accumulates in the fetal gut, starting at week 20 of gestation until birth. Intrauterine alcohol exposure can be detected in the meconium of a newborn as a third-trimester marker [36]. Within the first 2–24 h after birth, about 1 g of meconium was removed from the newborn and frozen at −80 °C for up to 30 months until analysis. The procedure for taking, storing, and determining the EtG samples/values in the meconium was described in Bakdash [37]. The positive detection limit for prenatal alcohol exposure varies slightly from study to study, but at least 10 ng/g EtG indicates alcohol exposure in the third trimester [37,38]. Therefore, the first cut-off value was set at 10 ng/g EtG. Other studies have found further effects with a higher cut-off [22,33,38]. Therefore, the second positive cut-off of EtG ≥ 112 ng/g was set to study the effects of higher levels of alcohol consumption within the sample and was defined in line with Grimm [22].

Maternal self-reports regarding alcohol consumption during pregnancy were recorded within the FRAMES study. In the third trimester, the women answered an interview question about their drinking behavior throughout pregnancy (no, I do not drink in general; no, I did not drink during pregnancy; yes, I rarely drank during pregnancy; yes, I drank one glass/day during pregnancy; and yes, I drank more than one glass/day during pregnancy) (category frequencies reported in [20]). Two groups were designed for data analysis based on the women’s self-reports: No drinking (‘I don’t drink in general’ + ‘I didn’t drink during pregnancy’) vs. drinking (‘I rarely drank during pregnancy’ + ‘I drank one glass/day during pregnancy’. No woman reported ‘Yes, I drank more than one glass day during pregnancy’). For the maternal self-report, all mothers (except for one mother) who reported prenatal alcohol consumption stated that they ‘rarely’ consumed alcohol [39]. Maternal self-reports during pregnancy on alcohol consumption behavior indicated a trend, with child meconium EtG for EtG ≥ 10 ng/g χ^2^(1) = 3.68, *p* = 0.055. However, this was not significantly correlated, with child meconium EtG for EtG ≥ 112 ng/g χ^2^(1) = 1.32, *p* = 0.251. 

Facial malformations: Photographic measurements of facial features were conducted analogously to Astley and colleagues with the FAS Facial Photographic Analysis Software (Version 2.1.0) [29] by placing a small paper sticker on the patient’s forehead between the eyebrows (see Figure 1; sticker size 12.70 mm × 19.05 mm). Two pictures were taken with a digital camera (Canon, Model IXUS 145, 14.0 Megapixels, Tokyo, Japan) while the study participants were in a seating position: One close up, with the patient’s head filling up the entire frame, and both ears being easily visible, and one frontal, with a three-fourths view. The facial expression of the child was always relaxed, with no smile, eyes wide open, and no glasses, and the hair was pinned back (Figure 1). All photographs were taken without the flashlight, and the camera was always secured on a tripod. The height of the tripod was dependent on the height of the subject. The camera lens had to align with the midface section of the subject’s head. All photographs were uploaded to the FAS Facial Photographic Analysis Software (Version 2.1.0) and analyzed following the program instructions. The palpebral fissure length (PFL), inner canthal distance (ICD), and thin upper lip (lip circularity) were measured by two independent raters using the computer mouse: The PFL by measuring the paper sticker and the distance between the exocanthion and endocanthion of the right eye and the left eye; the ICD by measuring the distance of the endocanthion of the right eye and the left eye; and the lip circularity by bordering the upper lip with the mouse. The program obtains a pixel value, a real length value (in millimeter), and an age- and sex-standardized *z*-score. The real length of the PFL value is calculated using the equation PFL (mm) = ((length of the sticker, mm/length of the sticker, pixel) × (PFL, pixel)) × 1.07 [30]. The real length of the ICD value was calculated using the equation ICD (mm) = ((length of sticker, mm/length of sticker, pixel) × (ICD, pixel)) [30]. For the PFL measurement, an adjustment value of 1.07 was added to the formula to adjust for the foreshortening effect. This is not necessary for the ICD measurement as it is recorded at the midline of the face. Therefore, no foreshortening effect is present [30,40]. The formula for the *z*-score was computed using the equation *PFL/ICD*
*z*-score = ((population mean *PFL/ICD* in mm)—(subject’s *PFL/ICD* in mm))/((population mean *PFL/ICD* in mm)—(population 1 SD *PFL/ICD* in mm)) [41]. The *z*-score was computed according to Hall (1989); all *z*-scores were age- and gender-dependent [42,43,44]. The underlying population was Caucasian. The upper lip was outlined with the computer mouse by each rater and was converted into the circularity value by the program. The circularity value was calculated using Circularity = (perimeter^2^/area) [45]. Higher PFL scores (i.e., longer eye length) decrease the chance of the child having FAS [46]. On the contrary, higher ICD scores (i.e., a larger distance between the two eyes) indicate a chance of the child having FAS, and higher lip circularity scores (i.e., smaller upper lip) increase the chance of the child suffering from FAS [30]. The described facial malformation criteria were rated by two independent and blind-trained raters; the interrater correlations for all three measures were highly significant, with *r* > 0.70 and *p <* 0.001 (Table 2), indicating the sufficient validity of the standardized rating method. For further analyses, the mean scores of both raters were calculated. Child lip circularity was significantly correlated with PFL (*r* = −0.203, *p <* 0.05), while child ICD was not associated. The PFL and ICD scores were independent (*r* = 0.091–0.068). The correlations between the length and *z*-scores for ICD (*r* = 0.999, *p <* 0.01) and PFL (*r* = 0.992, *p <* 0.01) were high. 

Cognitive Development: For the measurement of ‘fluid reasoning’ and ‘working memory’, we used the Wechsler Intelligence Test for Children (WISC V) [9], which is an individually administered test for children between 6 and 16 years of age. The test age- and sex-specific standard norms are based on a sample of about 1100 German children and adolescents from 2016. The WISC generates five primary standardized index scores (MW = 10, SD = 3). In the present study, the ‘Fluid Reasoning Index Score’ (‘Matrix Reasoning’ and ‘Figure Weights’ subtests; ability to detect an underlying conceptual relationship and to use reasoning to identify and apply rules) and the ‘Working Memory Index Score’ (‘Digit Span’ and the ‘Picture span’ subtests; ability to register, maintain, and manipulate visual and auditory information) were used to assess functional cognitive abnormalities. 

### 2.4. Confounders

We selected relevant confounders in concordance with theoretical considerations and earlier empirical results [2,23,26]. The following variables were tested in *t*-tests (Table 1, differences for EtG-positive versus -negative groups) and correlations (Table 3, with facial indices): Child’s sex; child’s age (years); childbirth weight (gram); current child’s weight (kg); current child’s height (cm); child’s current head circumference (cm); family socioeconomic status (combination of maternal and paternal education level (4-level: <9, 9, 10, or 13 years) and net family income (6-level: <1000 to >5000) during primary school age, sum-index theoretical range: 3–14); maternal age at delivery (years); and maternal smoking during pregnancy (yes (≥1 cigarette per day was interpreted as prenatal smoking)/no (≤1 cigarette per day was interpreted as no prenatal smoking)). EtG10+ and EtG10− children significantly differed in age and birth weight; child’s sex, birth weight, current weight, height, and head circumference measures were significantly correlated with facial characteristics. Afterward, these variables were controlled for in the ANCOVA analyses.

### 2.5. Statistical Analysis

Statistical analyses were carried out using IBM SPSS Statistics (Version 24.0, Armonk, NY: IBM Corp, 2016). Descriptive data are reported as means (M) and the standard deviation (SD). Uncontrolled mean differences were tested by *t*-tests and differences in frequencies by chi-squared tests (χ^2^), and are reported in descriptive data tables. Facial PFL, ICD, and circularity indices and continuous EtG scores were tested for normal distributions by the Shapiro–Wilk test in the total and subgroups. Lip circularity (total group: W = 0.945, *p <* 0.001) and continuous EtG data (total group: *W* = 0.297, *p <* 0.001) were not normally distributed. Therefore, we logarithmized (ln) the data. PFL (mm and Z), ICD (mm and Z), and circularity measures were tested in separate analyses of covariance (ANCOVA). Variables that were significantly (*p <* 0.05) correlated with at least one facial index or which significantly differentiated between EtG-positive versus -negative children within at least one cut-off group were controlled for in ANCOVA analyses. These criteria were applied to the following: Child’s age; childbirth weight; child’s sex; child’s weight; child’s height; and child’s head circumference (Table 1 and Table 3). The global analyses were run separately for the EtG10±, the EtG112±, and the self-report± groups. Post-hoc F-tests (univariate ANCOVA results) indicated significant group-differences. Facial markers, which were significantly predicted by prenatal variables in the foregoing ANCOVAs, were associated with cognitive ‘Fluid Reasoning’ and ‘Working Memory’ indices in confounder controlled partial correlations. The level of significance was defined as *p <* 0.05 (two-tailed) and *p <* 0.1 indicated a trend. 

## 3. Results

Table 4 reports the ANCOVA results. All global ANCOVA models for the prediction of a child’s facial features by alcohol consumption parameter and relevant confounders were significant (*p* < 0.05, Table 4). Post-hoc F-tests showed that affected children with EtG ≥ 10 ng/g with significance (*p =* 0.031, η_p_^2^ = 0.038) and with EtG levels ≥ 112 ng/g by trend (*p* = 0.055, η_p_^2^ = 0.030) displayed a smaller PFL than controls (Figure 2). The association was not significant for maternal self-reports (*p =* 0.164, η_p_^2^ = 0.016). Beyond the EtG status, the best predictors for child PFL were the child’s sex (*p =* 0.049, η_p_^2^ = 0.032), birth weight (*p =* 0.003, η_p_^2^ = 0.070), and current height (*p =* 0.004, η_p_^2^ = 0.066). Child ICD did not differ between exposed and non-exposed children (EtG10+: *p =* 0.777, η_p_^2^ = 0.001; EtG112+: *p =* 0.607, η_p_^2^ = 0.002). The only significant predictor was the child’s current head circumference (*p =* 0.037, η_p_^2^ = 0.036). PFL and ICD results were similar for FAS Facial Analysis Software Length- and *z*-scores. For child lip circularity, there was a significant difference between alcohol-exposed and non-exposed children for the 112 ng/g cut-off (*p =* 0.026, η_p_^2^ = 0.040; Figure 3). This effect was not found for the 10 ng/g cut-off (*p =* 0.340, η_p_^2^ = 0.008) or maternal self-report (*p =* 0.156, η_p_^2^ = 0.017). Lip circularity was also associated with the child’s current height (*p =* 0.043, η_p_^2^ = 0.034) and head circumference (*p =* 0.045, η_p_^2^ = 0.033).

Facial markers that were significantly predicted by child meconium EtG levels (PFL and lip circularity) were tested for functional relevance in association with cognitive outcomes. When controlling for relevant confounders in partial correlations (Table 5), affected children’s lip circularity was associated with their cognitive outcomes (Figure 4 and Figure 5): The fluid reasoning index reached significance within the EtG10+ group (*r*_p_ = −0.509, *p =* 0.031), while within the EtG112+ group, a medium correlation reached no significance (*r*_p_ = −0.391, *p =* 0.298). When interpreting the effect size measure, lip circularity correlations with a high effect (*r* > 0.500) were, within the EtG112+ group, present for the working memory index (a statistical trend was also seen in the EtG10+ group: *r* = −0.418, *p =* 0.084). There was no statistical significance for these correlations. The PFL scores of EtG+ children were not significantly correlated with cognitive outcomes, and correlation coefficients were low (*r* < 0.129).

## 4. Discussion

The present study examined the effects of PAE on structural facial changes and associated impairments in cognitive functions in young adolescents. The meconium biomarker EtG and maternal self-reports during the third trimester were used to identify prenatal alcohol exposure during pregnancy. For both cut-offs (10 ng/g was statistically significant, and 112 ng/g was significant by trend), exposed children had a smaller PFL value. For child lip circularity, the 112 ng/g cut-off was significant. Maternal self-reports did not show any statistical significance. This may be associated with the inaccuracy of the self-report, as discussed in previous papers [20,46].

Furthermore, it has been reported that mothers who consumed a higher amount of alcohol during pregnancy are the most likely to answer the self-report inaccurately, thus skewing the results [33]. Earlier research has already shown that maternal self-reports are inclined to underreport PAE: In their review, Lange and colleagues reported that underestimation was four-times the actual value when using maternal self-reports in comparison to meconium biomarker data [47]. One study found cognitive outcomes to be better predicted by the meconium biomarker than by maternal self-reports during pregnancy [16]. Furthermore, predictive quality comparisons have been missing until now. The present study contributes results for the facial phenotype. The PFL and the lip circularity measure are two of the three common facial features employed for FAS diagnosis. As expected, the EtG biomarker was associated with PFL and lip circularity. The ICD—not listed as a facial feature in the FAS diagnostic guidelines—was of negligible relevance. We observed that only the head circumference of the patient determines the ICD. The greater the head circumference of the patient, the greater the distance between the patient’s eyes, and thus the greater the ICD. 

The results lend validity to the EtG meconium marker and the harm of prenatal alcohol consumption for child development. In detail, these results found support for the notion that a lower consumption of alcohol during pregnancy (EtG10+) has a greater impact on a shortened PFL length. Furthermore, it was observed that greater amounts of alcohol consumption (EtG112+) had a considerable impact on lip circularity, resulting in a smaller upper lip. Muggli [2] suggests that certain FAS features might only develop at higher levels of alcohol consumption. A baby’s face already starts to develop at day 17–18 after fertilization [48]. The neural crest development is affected by ethanol during all stages, which causes facial malformations in the baby [27]. In mice studies, it has been observed that the timing of prenatal alcohol consumption greatly affects the facial changes in the fetus [27]. It was detected that alcohol exposure to the fetus at gestational day 7, which is equivalent to gestational day 15–17 in humans [49], causes changes in the midface, specifically, a smooth philtrum and a stretched upper lip [27]. This can be compared to exposure at gestational day 8.5, which only mildly influences the midface area and philtrum, but causes a shorter upper lip [27]. Previous studies also display these variabilities in facial malformations with heavy PAE [25,48,50]. Additionally, factors such as oxidative stress, growth factor involvement, epigenetic influences, and a decreased biosynthesis of retinoic acid might also influence the development of the infant’s face [2,51]. Despite these diverse influencing factors, the significant association between the EtG biomarker and the child’s facial characteristics is impressive and indicates a stable association.

The question of how much alcohol was consumed during pregnancy when giving birth to an EtG-positive child is difficult to answer. Goecke [33] reported that 20.4% of their study sample had a positive EtG marker [33]. In contrast, only 0.77% of the general population are diagnosed with FASD [7]. The high prevalence of positive EtG compared to the lower prevalence of diagnosed FASD children might be an indicator that not every child has symptomatic FASD or may be diagnosed falsely. There is a high correlation between FASD and symptomatic attention deficit hyperactivity disorder (ADHS) [52]. Seventy percent of children who suffer from FASD present ADHS symptoms [52]. Furthermore, the EtG biomarker indicates the amount of ethanol metabolite that was present in meconium, but we cannot precisely reflect how much of this originated from the consumption of alcohol or through specific food (e.g., fresh fruit juice and fresh fruit) [53,54]. Therefore, positive EtG cases might not be clinically and pathologically relevant. The consumption of alcohol during pregnancy impacts the child negatively, but it does not necessarily lead to diagnostic FASD [55]; the alcohol-induced vulnerability of the child cumulates with further risk factors, increasing the chance of FASD [56].

For example, a low socioeconomic background increases the vulnerability of developing diagnostic FASD [57]. Most (70.9%) of the children born to prenatally alcohol-consuming mothers from a low socioeconomic background develop FASD, whereas, in children born to mothers from a high socioeconomic background, only 4.5% develop FASD [57,58].

Until now, few studies have attempted to link meconium EtG levels with concrete prenatal drinking amounts. In contrast, there are results for other meconium ethanol metabolites: Mothers of fatty acid ethyl esters (FAEE)—EtG and FAEE are highly correlated with each other [53]. Mothers of EtG and FAEE-positive children drank an average of 10 drinks/week and no fewer than four drinks/week [59]. One or two drinks per day were not enough to find positive FAEE meconium [60]. A particularly crucial point in this context is that the facial phenotype forms during early pregnancy [28], whereas the EtG meconium levels only indicate the intrauterine alcohol exposure during the last trimester of pregnancy [37]. Therefore, we can only estimate how much alcohol was consumed over the course of the pregnancy. Other research suggests that most women reduce their alcohol intake once they find out that they are pregnant [61]. Proportions, as well as quantities, of drinking, usually decrease during the course of pregnancy [62]. An additional problem may be that some women consumed alcohol in the first weeks of pregnancy, but stopped the consumption of alcohol before the third trimester, thus indicating no EtG meconium levels. However, a recent study investigated the prenatal drinking patterns of 19 mothers of FASD children. They concluded that only 10.5% of the mothers continued drinking after pregnancy recognition—based on their self-reports [63]. Nonetheless, both variables are associated, consistent with the hypothesis that mothers who consumed alcohol in late pregnancy were already consuming alcohol in early pregnancy. A recent study confirmed the assumption by identifying a ‘low-to-moderate PAE sustained across gestation’ and a ‘high PAE sustained across gestation’ type with the highest effects on child development [24].

Our findings do not stop at the point that prenatal alcohol consumption is associated with a child biomarker. An additional focus was placed on the functional relevance of this association. Within the EtG-positive groups, the lip circularity measure was correlated with child fluid reasoning and working memory competence. All correlations between lip circularity measures and cognitive functioning were *r* > 0.30 for practical relevance, with partial significance. Research has shown that the brain is severely impacted when the mother consumes alcohol during pregnancy [64]. Children with FASD perform more poorly on IQ tests, compared to unaffected children, especially in terms of verbal learning, working memory, and fluid intelligence [65,66]. The brain and the face develop simultaneously during pregnancy. Therefore, a correlation between brain impairments and facial changes is not unexpected. The lip circularity was superior to the PFL measure in predicting cognitive impairments, meaning a thinner upper lip was the better criterion for cognitive impairment in young adolescents. These results are important for demonstrating the developmental consequences of early prenatal alcohol exposure and validating the lip circularity measure as a facial FAS diagnosis criterion, but must be interpreted carefully to avoid stigmatization. At the same time, when comparing the interrater-accuracy of the different values, it was found that the lip circularity had the greatest inaccuracy.

The ICD and PFL values were calculated by the FAS Facial Photographic Analysis software by setting points on the photograph. In contrast, the lip circularity value was enumerated by outlining the lip with a computer mouse. Lining the lip with a computer mouse might cause a greater inaccuracy between the different independent-raters as it can be difficult for some raters to trace the lip accurately. This method might be prone to more errors, rather than just that associated with setting points on the photograph. This could be the reason why the lip circularity exhibited a greater inaccuracy for the different independent-raters. Therefore, it is of great importance that the same individual trains each independent-rater extensively beforehand, to ensure highly accurate results. Conversely, in the research paper by Astley, only the PFL measurements were extensively examined, and not the lip circularity measurements [42].

The present study is the first study to validate FAS facial diagnosis software as a potential digital tool for clinical practice in a German general population sample. The digitally determined FAS facial features were significantly correlated with PAE measures and developmental outcomes. Even in young adolescents, when facial characteristics begin to decrease and become less clear [14,29], standardized *z*-scores were as reliable as absolute length measures in millimeters. However, the most critical finding was that the association of meconium EtG and facial features only became significant when relevant covariates were controlled for, including the child’s age, sex, weight, height, and head circumference. The actual PFL *z*-scores are based on samples of American children, taking age and gender into account [44]; lip circularity measures are not standardized. When using the FAS Facial Photographic Analysis software to measure the facial changes in patients, it is important to study confounders beforehand and to include them in the analysis. There is a need to expand existing standard norms—minimally by weight, height, and head circumference specifications—before the software can be systematically used in clinical settings. In future studies, including these confounders in the *z*-score calculation might be beneficial as the factors height, weight, and head circumference are correlated with the eye size [67,68]. Furthermore, future studies should try to replicate the research in larger cohorts. Additionally, when replicating this research in a larger cohort, the sexes should be separated in the analyses. 

## 5. Conclusions

This is the first study to use meconium EtG to test the hypothesis that subliminal PAE has a detectable—if even subclinical—effect on craniofacial structures, enduring into early adolescence. The results suggest that prenatal alcohol consumption has a sustained impact on craniofacial development: When including relevant confounders (age, sex, height, weight, and head circumference), we observed that the PFL and lip circularity were affected in young adolescence children. These characteristics are associated with the FAS Diagnostic Guidelines. In this context, the EtG meconium biomarker is a more valid predictor than maternal self-reports. The present study not only shows the consequences of intrauterine alcohol exposure on the development of the face, but also displays the correlation between facial and cognitive changes. We conclude that the FAS Facial Photographic Analysis software can be used in clinical settings to reduce measurement errors in lip-philtrum-guide and ruler use by well-trained independent raters, following future studies on adequate standardized norms. Overall, it can be concluded that prenatal alcohol consumption is a major risk factor for the unborn child, accompanied by multi-level changes. Therefore, avoiding alcohol during pregnancy is the safest option.

## Figures and Tables

**Figure 1 brainsci-11-00154-f001:**
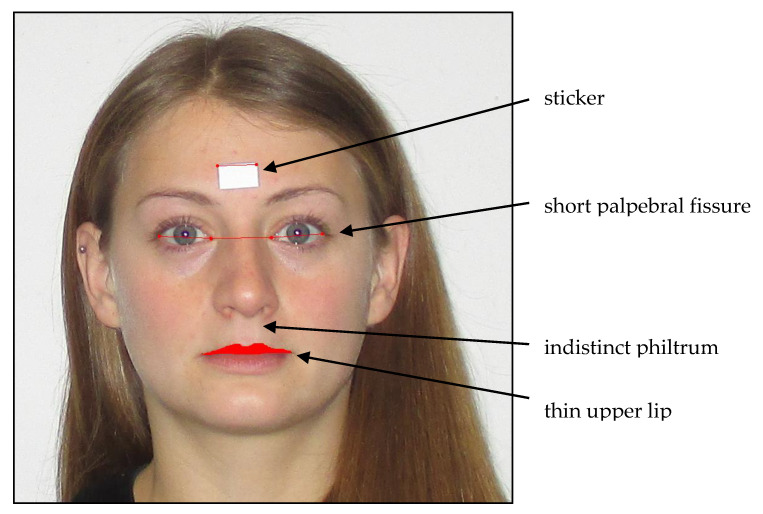
Typical facial abnormalities in fetal alcohol syndrome (FAS) children and FAS software measures (modeled in a non-affected adult person).

**Figure 2 brainsci-11-00154-f002:**
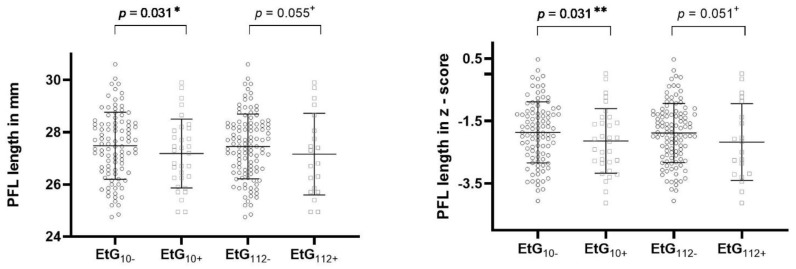
PFL length in millimeters and *z*-score for EtG 10 ng/g (EtG10+ *n* = 32) and 112 ng/g (EtG112+ *n* = 20) cut-off values. Higher PFL value = longer eyes. Univariate ANCOVA results: + *p* < 0.01, * *p <* 0.05, and ** *p <* 0.01.

**Figure 3 brainsci-11-00154-f003:**
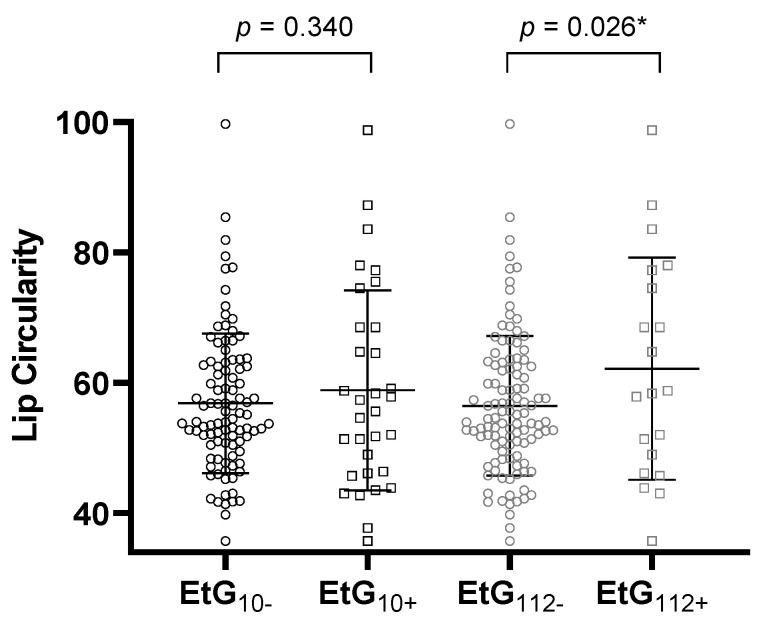
Lip circularity for the EtG 10 ng/g (EtG10+ *n* = 32) and 112 ng/g (EtG112+ *n* = 20) cut-offs. Higher lip circularity value = smaller upper lip size. Univariate ANCOVA results: * *p <* 0.05

**Figure 4 brainsci-11-00154-f004:**
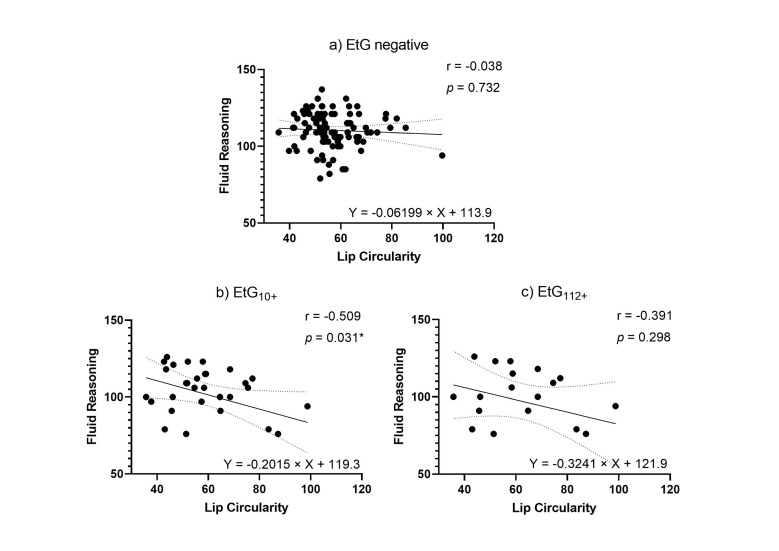
(**a**) Lip circularity versus fluid reasoning for EtG-negative individuals (10 ng/g cut-off, EtG-negative *n* = 97). (**b**) Lip circularity versus fluid reasoning for EtG10+ individuals (EtG_10_+ *n* = 32). (**c**) Lip circularity versus fluid reasoning for the EtG112+ (EtG_112_+ *n* = 20) cut-off. Higher lip circularity value = smaller upper lip size. * *p <* 0.05.

**Figure 5 brainsci-11-00154-f005:**
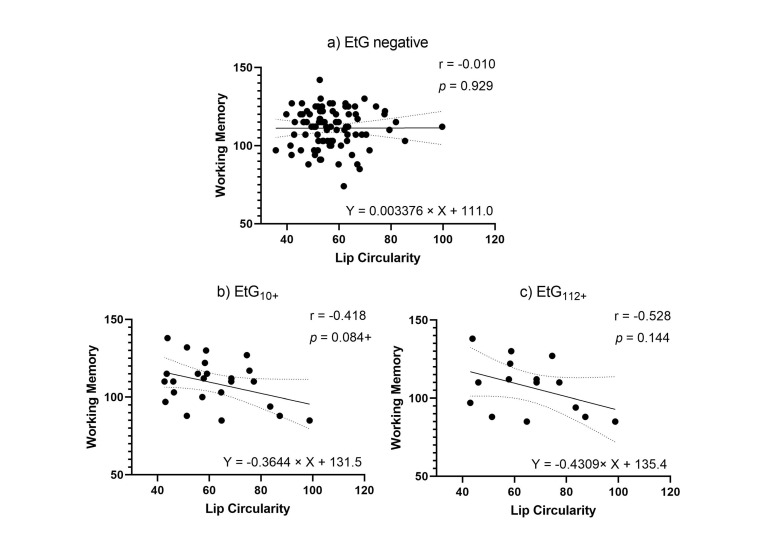
(**a**) Lip circularity versus working memory for EtG-negative individuals (the 10 ng/g cut-off, EtG-negative *n* = 97). (**b**) Lip circularity versus fluid reasoning for EtG10+ individuals (EtG_10_+ *n* = 32). (**c**) Lip circularity versus working memory for EtG112+ (EtG_112_+ *n* = 20) cut-offs. Higher lip circularity value = smaller upper lip size. ^+^
*p <* 0.10.

**Table 1 brainsci-11-00154-t001:** Frequency (*n*), means (M), and standard deviations (SD) of sample characteristics.

	Total	EtG_10_−	EtG_10_+	Statistics			EtG_112_−	EtG_112_+	Statistics			Self−	Self+	Statistics		
*n:*	129	97	32	*t(df) ^a,c^*	*p*	d	109	20	*t(df) ^a,c^*	*p*	d	97	32	*t(df) ^a,c^*	*p*	d
Prenatal:																
EtG (ng/g)	82.7 (301.2)	--	333.4 (537.0)	--	--	--	5.28 (17.4)	504.7 (622.7)	−19.0 (127)	0.000 **	1.13	92.5 (342.6)	52.9 (99.4)	−1.44 (127)	0.153	0.16
sex (m/f) ^a^	66/63	51/46	15/17	0.31	0.576	0.49	57/52	9/11	0.36	0.549	0.53	46/51	20/12	2.19	0.139	0.13
birth weight (grams)	3439 (450.9)	3382 (411.8)	3613 (522.2)	−2.28 (44.4)	0.027 *	0.49	3408 (441.7)	3612 (473.0)	−1.88 (127)	0.062 ^+^	0.45	3393 (451.8)	3579 (424.9)	−2.05 (127)	0.042 *	0.42
maternal age at delivery (years)	32.7 (4.53)	32.5 (4.73)	33.1 (3.92)	−0.63 (127)	0.533	0.14	32.7 (4.63)	32.6 (4.06)	0.15 (127)	0.881	0.02	32.1 (4.61)	34.5 (3.79)	−2.67 (127)	0.009 **	0.57
maternal smoking pregnancy ^d^	0.65 (2.31)	0.60 (2.21)	0.81 (2.63)	−0.45 (127)	0.651	0.19	0.53 (2.09)	1.30 (3.26)	−1.37 (127)	0.173	0.28	0.60 (2.21)	0.81 (2.63)	−0.45 (127)	0.651	0.09
Today:																
age (years)	13.30 (0.32)	13.2 (0.27)	13.4 (0.39)	−2.47 (41.3)	0.018 *	0.60	13.20 (0.29)	13.41 (0.42)	−1.91 (22.3)	0.068 ^+^	0.58	13.31 (0.31)	13.29 (0.34)	−0.77 (127)	0.44	0.06
weight (kg)	50.9 (10.2)	49.94 (9.61)	53.64 (11.38)	−1.81 (127)	0.073 ^+^	0.35	50.2 (9.95)	54.7 (10.7)	−1.84 (127)	0.068 ^+^	0.44	50.8 (10.3)	51.1 (9.95)	−0.13 (127)	0.896	0.03
height (cm)	161.4 (7.83)	161.1 (7.70)	162.24 (8.32)	−0.70 (127)	0.487	0.14	161.0 (7.57)	163.4 (9.12)	−1.22 (127)	0.226	0.29	161.6 (7.63)	160.7 (8.49)	0.56 (127)	0.575	0.11
head (cm)	55.0 (1.68)	54.87 (1.77)	55.23 (1.37)	−1.07 (127)	0.285	0.23	54.9 (1.72)	55.3 (1.47)	−0.87 (127)	0.387	0.25	54.84 (1.64)	55.32 (1.77)	−1.40 (127)	0.162	0.28
family status ^b^	11.5 (2.14)	11.39 (2.26)	11.8 (1.74)	−0.89 (127)	0.374	0.20	11.47 (2.19)	11.6 (1.90)	−0.25 (127)	0.801	0.09	11.4 (2.17)	11.9 (2.02)	−1.27 (127)	0.204	0.24
fluid reasoning	108.2 (14.8)	110.4 (11.7)	101.7 (20.3)	2.23 (37.0)	0.032 *	0.61	110.2 (11.3)	97.0 (24.0)	2.37 (19.5)	0.028 *	0.70	108.4 (15.3)	107.5 (12.8)	0.27 (120)	0.786	0.06
working memory	110.7 (12.8)	111.2 (12.3)	109.1 (14.8)	0.70 (112)	0.484	0.15	111.3 (12.1)	107.2 (17.2)	0.88 (16.1)	0.392	0.28	111.2 (12.2)	109.3 (14.8)	0.66 (112)	0.512	0.14
PFL Len (mm)	27.4 (1.29)	27.5 (1.28)	27.2 (1.32)	1.12 (127)	0.264	0.23	27.45 (1.24)	27.16 (1.57)	0.94 (127)	0.351	0.20	27.45 (1.37)	27.28 (1.02)	0.74 (70.6)	0.464	0.14
Z	−1.93 (1.00)	−1.86 (0.98)	−2.14 (1.03)	1.38 (127)	0.172	0.28	−1.88 (0.95)	−2.17 (1.23)	1.19 (127)	0.237	0.26	−1.90 (1.06)	−2.03 (0.80)	0.80 (68.8)	0.429	0.14
ICD Len (mm)	31.9 (2.47)	31.76 (2.57)	32.2 (2.16)	−0.90 (127)	0.371	0.19	31.8 (2.52)	32.4 (2.17)	−0.98 (127)	0.321	0.26	31.6 (2.35)	32.7 (2.68)	−2.20 (127)	0.029 *	0.44
Z	0.40 (1.03)	0.35 (1.07)	0.55 (1.00)	−0.93 (127)	0.354	0.19	0.34 (1.05)	0.62 (0.88)	−1.05 (127)	0.297	0.29	0.29 (0.98)	0.74 (1.11)	−2.18 (127)	−0.451	0.43
lip circularity	57.4 (12.0)	56.9 (10.7)	58.9 (15.3)	−0.69 (41.5)	0.416	0.15	56.5 (10.7)	62.2 (17.0)	−1.44 (21.8)	0.164	0.40	58.0 (11.7)	55.3 (12.9)	1.12 (127)	0.263	0.22

Notes. ^a^ ‘Child sex’ statistic: Chi-squared χ^2^(1) test. EtG: Ethyl glucuronide. ^b^ Socioeconomic status: FRANCES I, combination of maternal/paternal education level (4-level: <9, 9, 10, or 13 years) and net family income (6-level: <1000 to >5000) (sum-index, theoretical range: 3–14). ^c^ School types represent ranked ordinal data. Therefore, a Mann–Whitney U-test was performed. ^d^ Cigarettes/day. Child fluid reasoning and working memory: WISC V, theoretical range: MW = 100, SD = 15 [1]. PFL: palpebral fissure length; the elliptic space between the medial and lateral canthi of the two open lids. Higher PFL value = longer eyes. ICD = inner canthal distance; distance between the two medial canthi of the eyes. Higher ICD value = increased distance between the eyes. Circularity = thinness of the upper lip. Higher Circularity value = smaller upper lip size. Len = true length in mm. Z = calculated *z*-score, according to Hall (1989) [44]; all *z*-scores are age- and gender-dependent and calculated by the program. χ^2^: *df* = x; *t: df* = x. ^+^
*p <* 0.10, * *p <* 0.05, and ** *p* < 0.01.

**Table 2 brainsci-11-00154-t002:** FAS software score correlations (Pearson’s r) between two independent raters, *n* = 129.

			Rater 1				
			PFL		ICD		Circularity
			Length	*z*-score	Length	*z*-score	
Rater 2	PFL ^a^	Length	0.884 **	0.887 **	0.127	0.128	−0.090
		*z*-score	0.869 **	0.876 **	0.097	0.098	−0.079
	ICD	Length	0.047	0.039	0.970 **	0.970 **	−0.132
		*z*-score	0.037	0.029	0.964 **	0.965 **	−0.120
	Circularity		−0.195 *	−0.186 *	−0.036	−0.037	0.727 **

Notes. ^a^ Left-Right-Mean Score. Length in mm. *z*-score (see [43,44]): Age and gender norms. PFL = palpebral fissure length; the elliptic space between the medial and lateral canthi of the two open lids. Higher PFL value = longer eyes. ICD = inner canthal distance; distance between the two medial canthi of the eyes. Higher ICD value = increased distance between the eyes. Circularity = thinness of the upper lip. Higher circularity value = smaller upper lip size. Length = true length in mm, calculated by the program. Z = *z*-score calculated by the program. * *p <* 0.05, and ** *p* < 0.01.

**Table 3 brainsci-11-00154-t003:** Testing potential confounders in Pearson’s correlations and *t*-tests (r/t(*p*)), *n* = 129.

	PFL		ICD		Circularity ^a^
	Len	Z	Len	Z	
EtG ^a^	−0.060 (0.500)	−0.083 (0.350)	0.086 (0.335)	0.089 (0.313)	0.085 (0.337)
sex (male/female)	3.09 (0.002) **	3.00 (0.002) **	2.60 (0.011) *	2.51 (0.013) *	−1.55 (0.124)
age (years)	0.113 (0.200)	0.038 0(.669)	0.091 (0.305)	0.080 (0.365)	−0.169 (0.055) ^+^
birth weight (grams)	0.308 (0.000) **	0.293 (0.001) **	0.253 (0.004) **	0.247 (0.005) **	−0.054 (.544)
weight (grams)	0.256 (0.003) **	0.223 (0.011) *	0.204 (0.020) *	0.202 (0.022) *	−0.162 (0.066) ^+^
height (cm)	0.382 (0.000) **	0.349 (0.000) **	0.203 (0.021) *	0.198 (0.025) *	−0.239 (0.006) **
head (cm)	0.300 (0.001) **	0.284 (0.001) **	0.348 (0.000) **	0.338 (0.000) **	−0.008 (0.925)
family status	−0.056 (0.531)	−0.052 (0.556)	−0.154 (0.081) ^+^	−0.162 (0.066) ^+^	−0.059 (0.503)
maternal age at delivery	−0.067 (0.450)	−0.075 (0.396)	0.069 (0.439)	0.070 (0.431)	0.088 (0.320)
maternal smoking pregnancy	0.021 (0.816)	0.022 (0.802)	0.005 (0.959)	0.004 (0.965)	0.032 (0.721)

Notes. ^a^ ln(x + 1) transformed continuous EtG data, and ln(x) transformed lip circularity data. Socioeconomic status: Combination of maternal/paternal education level (4-level: <9, 9, 10, or 13 years) and net family income (6-level: <1000 to >5000) (sum-index, theoretical range: 3–14). t: *df* = 127. Maternal smoking during pregnancy: yes = smoking ≥ 1 cigarette per day. PFL = palpebral fissure length; the elliptic space between the medial and lateral canthi of the two open lids Higher PFL value = longer eyes. ICD = inner canthal distance; distance between the two medial canthi of the eyes. Higher ICD value = increased distance between the eyes. Circularity = thinness of the upper lip. Higher circularity value = smaller upper lip size. Len = true length. Z = calculated *z*-score, according to Hall (1989) [44], where all *z*-scores are age-dependent and calculated by the program. ^+^
*p <* 0.10, * *p <* 0.05, and ** *p <* 0.01.

**Table 4 brainsci-11-00154-t004:** EtG-associated differences in facial parameters (ANCOVAs, *n* = 129, df: 7/121).

	EtG10			EtG112			Self-Report		
	F	*p*	η_p_^2^	F	*p*	η_p_^2^	F	*p*	η_p_^2^
PFL Length	6.72	0.000 **	0.280	6.53	0.000 **	0.274	6.18	0.000 **	0.263
EtG/Self	4.79	0.031 *	0.038	3.76	0.055 ^+^	0.030	1.96	0.164	0.016
sex	3.94	0.049 *	0.032	4.09	0.045*	0.033	5.16	0.025 *	0.041
age	0.06	0.806	0.001	0.01	0.916	0.000	0.02	0.888	0.000
birth weight	9.15	0.003 **	0.070	8.30	0.005 **	0.064	7.58	0.007 **	0.059
height	8.59	0.004 **	0.066	9.55	0.002 **	0.073	8.49	0.004 **	0.066
weight	0.07	0.799	0.001	0.32	0.895	0.000	0.00	0.960	0.000
head circumference	0.14	0.711	0.001	0.12	0.733	0.001	0.25	0.619	0.002
PFL *z*-score	5.99	0.000 **	0.257	5.83	0.000 **	0.252	5.46	0.000 **	0.240
EtG/Self	4.76	0.031 *	0.038	3.88	0.051 ^+^	0.031	1.89	0.172	0.015
sex	3.51	0.063 ^+^	0.028	3.65	0.058 ^+^	0.029	4.66	0.033 *	0.037
age	0.35	0.558	0.003	0.53	0.469	0.004	0.97	0.327	0.008
birth weight	8.56	0.004 **	0.066	7.77	0.006 **	0.060	7.03	0.009 **	0.055
height	7.80	0.006 **	0.061	8.72	0.004 **	0.067	7.73	0.006 **	0.060
weight	0.02	0.887	0.000	0.01	0.935	0.000	0.00	0.961	0.000
head circumference	0.23	0.663	0.002	0.20	0.656	0.002	0.36	0.549	0.003
ICD Len	3.35	0.003 **	0.162	3.38	0.002 **	0.164	3.75	0.001 **	0.178
EtG/Self	0.08	0.777	0.001	0.27	0.607	0.002	2.43	0.122	0.020
sex	2.06	0.154	0.017	2.11	0.149	0.017	1.70	0.195	0.014
age	0.01	0.904	0.000	0.02	0.883	0.000	0.02	0.881	0.000
birth weight	2.25	0.136	0.018	2.24	0.137	0.018	1.95	0.165	0.016
height	0.16	0.686	0.001	0.15	0.697	0.001	0.32	0.575	0.003
weight	0.23	0.633	0.002	0.22	0.641	0.002	0.22	0.638	0.002
head circumference	4.47	0.037 *	0.036	4.52	0.036 *	0.036	4.02	0.047 *	0.032
ICD *z*-score	3.17	0.004 **	0.155	3.21	0.004 **	0.157	3.56	0.002 **	0.171
EtG/Self	0.12	0.726	0.001	0.35	0.554	0.003	2.41	0.123	0.020
sex	1.84	0.166	0.016	1.99	0.161	0.016	1.58	0.212	0.013
age	0.06	0.808	0.000	0.07	0.788	0.001	0.07	0.793	0.001
birth weight	2.16	0.145	0.018	2.16	0.145	0.018	1.90	0.171	0.015
height	0.15	0.701	0.001	0.13	0.716	0.001	0.28	0.594	0.002
weight	0.27	0.605	0.002	0.26	0.612	0.002	0.27	0.604	0.002
head circumference	4.14	0.044 *	0.033	4.20	0.043 *	0.034	3.71	0.056 *	0.030
Circularity	2.31	0.031 *	0.118	2.98	0.006 **	0.147	2.49	0.020 *	0.126
EtG/Self	0.92	0.340	0.008	5.09	0.026 *	0.040	2.04	0.156	0.017
sex	3.12	0.080 ^+^	0.025	2.92	0.090 ^+^	0.024	3.08	0.082 ^+^	0.025
age	2.25	0.136	0.018	2.92	0.090 ^+^	0.024	1.56	0.214	0.13
birth weight	0.26	0.661	0.002	0.44	0.506	0.004	0.02	0.896	0.000
height	3.83	0.053 ^+^	0.031	4.20	0.043 *	0.034	4.86	0.029 *	0.039
weight	0.37	0.543	0.003	0.50	0.479	0.004	0.22	0.644	0.002
head circumference	3.77	0.055 ^+^	0.003	4.10	0.045 *	0.033	4.17	0.043 *	0.033

Notes. N: EtG10− (97) vs. EtG10+ (32); EtG112− (109) vs. EtG112+ (20); Self− (97) vs. Self+ (32). All analyses controlled for the child’s age, childbirth weight, child’s sex, child’s weight, child’s height, and child’s head circumference as covariates. EtG = ethyl glucuronide. PFL = palpebral fissure length; the elliptic space between the medial and lateral canthi of the two open lids. Higher PFL value = longer eyes. ICD = inner canthal distance; distance between the two medial canthi of the eyes. Higher ICD value = increased distance between the eyes. Circularity = thinness of the upper lip. Higher circularity value = smaller upper lip size. Len = true length- Z = calculated *z*-score by program. ^+^
*p <* 0.10, * *p <* 0.05, and ** *p <* 0.01.

**Table 5 brainsci-11-00154-t005:** Partial correlations (r p) of child PFL and lip circularity with the fluid reasoning and working memory index within the affected EtG+ groups.

	*n*	Fluid Reasoning Index	Working Memory Index
EtG_10_+	32 (*df* = 16)		
PFL len		0.113 (0.654)	0.058 (0.818)
PFL *z*-score		0.117 (0.644)	0.044 (0.861)
Circularity		−0.509 (0.031 *)	−0.418 (0.084 ^+^)
EtG_112_+	20 (*df* = 7)		
PFL len		−0.010 (0.980)	−0.129 (0.741)
PFL *z*-score		0.007 (0.986)	−0.129 (0.741)
Circularity		−0.391 (0.298)	−0.528 (0.144)

Notes. All correlations controlled for the child’s age, childbirth weight, child’s sex, child’s weight, child’s height, and child’s head circumference as covariates. EtG = ethyl glucuronide. PFL = palpebral fissure length; the elliptic space between the medial and lateral canthi of the two open lids. Higher PFL value = longer eyes. ICD = inner canthal distance; distance between the two medial canthi of the eyes. Higher ICD value = increased distance between the eyes. Circularity = thinness of the upper lip. Higher circularity value = smaller upper lip size. Len = true length in mm. Z = calculated *z*-score by program. ^+^
*p <* 0.10, * *p <* 0.05.

## Data Availability

The data presented in this study are available on request from the corresponding author.

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
