# Peer review of "Prenatal Alcohol Exposure and the Facial Phenotype in Adolescents: A Study Based on Meconium Ethyl Glucuronide"

_brainsci, 2021, doi:10.3390/brainsci11020154_

Round 1

Reviewer 1 Report

This paper describes outcomes in a German population focusing on ethyl glucuronide (EtG), neurocognitive development, and facial characteristics of FASDs in adolescents. Overall, the results confirm what has been described in other populations in other parts of the world, however, few of those studies assess adolescents. Findings are somewhat modest, likely in part due to small sample sizes in the two EtG groups (n=32 and n=20). The text requires some clarification as noted below. One major point is that EtG only reflects recent drinking, this is briefly mentioned in the Discussion but the authors need to consider that there may be people in the control group who were exposed early in pregnancy around the time of gastrulation and that this could contribute to the variability in that group.

It’s not clear what is meant by the statement on line 92 - “Animal studies report, that the facial shape is formed until day 16 of pregnancy [30], representing an early marker of intrauterine alcohol exposure” or on line 98 – “There is rare evidence, which subclinical effects alcohol consumption can have on facial characteristics of a child.”

Since p<0.1 is considered a trend for most outcomes in the paper, it should be considered a trend in Line 234 - “Maternal self-report during pregnancy on alcohol consumption behavior was not significantly correlated with child meconium EtG for both cut-offs: EtG ≥ 10 ng/g χ2(1) = 3.68, p = .055, EtG ≥ 112 ng/g χ2(1) = 1.32, p = 236 .251.”

Figures 4 and 5 are confusing because of the decision to exclude some data points from the graphs – did the analyses for the EtG10+ group include the 20 people in the EtG112+ group? If so, this needs to be clear. Also, please add p values to graphs.

Line 123 – “Until now, there are two studies, which associated self-report operationalized alcohol exposure with facial anomalies in infants.” Please add references to the papers described.

Line 341 – chi-squared test, not Qui

Line 360 - should include that the authors are using p<0.1 as a trend

Line 440 – pervious should be previous

Line 466 – neuronal crest should be neural crest

Line 490 - define ADHS

Sentence beginning on line 555 does not seem like the start of a new paragraph.

Tables 1 & 3 need to be reformatted – consider presenting in landscape orientation.

Consider using bold font in Tables 1, 3 and 4 to make clear the significant outcomes

Reviewer 2 Report

Review of brainsci-1056218

Title----Prenatal alcohol exposure and facial phenotype in adolescents - a study based on meconium ethyl glucuronide.

One problem with accurate identification of children with FASD has always been the problem of maternal self-report.  While there are clear diagnostic criteria for the more severe FAS, cognitive-behavioral issues related to prenatal alcohol exposure  sometimes looks very similar to other developmental disorders, such as ADHD.  The addition of a biomarker, as found through meconium, may be very helpful in attaining proper therapeutics for children who suffer from FASD.

Methodology:  One this I find unusual is how the groups were set by the MEC level.  I understand that this metabolite is only present in the mec as s result of maternal drinking, and that the two levels, 10 and 112ng/g make sense in terms of low versus high exposure.  However, the confusing part is this: the author designates 4 groups, basically <10 negative and a >10 positive group.  Already I find this problematic as a value, then, of 9.8 would be considered negative which it clearly indicates exposure. Moreover, this problem increase in the 112 groups.  The authors for this analysis compare cases that had MEC >112 as positive (which makes sense) but then <112 as negative when this is a very high number to consider negative.  A better way would be to have three groups: 0-5 ng/g as negative, then 10+ (with a set upper limit) as low positive and 112+ as a high positive. 

Results: One big problem that stands out right away is that in table 3 (confounders) the authors show significant sex differences.  Effects of PAE by sex are now well known and even without these data showing such significance, the authors should have separated the sexes in subsequent analyses.

Also, back to the issue of the negative ‘control’ groups as described above under methology, I cant understand why the plotted data points for the two negative conditions (<10 and <112) are exactly the same points? If the 112- is <112 then that would also incorporate much of the 10+ data.  So perhaps they are using one single negative control group but then labelling as difference groups (10- and 112-) with methods descriptions as less than 10 and less than 112? If that is the case and the two controls groups are the same (and both are below 10 then that needs to be cleared up.

Minor:  There are = grammatical errors throughout. Line 27: implemented should be replaced with assessed. Line 29: cut off should be plural.  The sentences in the abstract feel abrupt and short and would flow better if combined.  Line 53: (7.8) is repeated. Line 55-57: Alcohol ‘is’….’that’ can damage. Line 57-59: sentence fragment.  As I continue to read on, these writing issues are persistent and it would take hours and hours for me to highlight every mistake in grammar, word choice or sentence structure.  I suggest the authors use a professional English editing service to improve the writing.

Round 2

Reviewer 2 Report

This manuscript is greatly improved form the original form.  I reviewed my review critiques as well as those from the additional reviewer and I feel that both of our queries and issues were adequately resolved.  I do want to mention that FASD is Fetal Alcohol Spectrum Disorders (and in some places the authors have spelled it without the s on disorders (or singular).   This should be corrected.   In all the English language was improved and clarifications were made as well as some correction.  My scores are as such simply because of the sex effect and how the authors are unable to secure enough subjects to give power to a sex by sex analysis.  However, as long as the caveat is clarified, it should be fine.